# Four New Wood-Inhabiting Fungal Species of Peniophoraceae (Russulales, Basidiomycota) from the Yunnan-Guizhou Plateau, China

**DOI:** 10.3390/jof8111227

**Published:** 2022-11-21

**Authors:** Lei Zou, Xiaolu Zhang, Yinglian Deng, Changlin Zhao

**Affiliations:** 1Key Laboratory for Forest Resources Conservation and Utilization in the Southwest Mountains of China, Ministry of Education, Southwest Forestry University, Kunming 650224, China; 2College of Biodiversity Conservation, Southwest Forestry University, Kunming 650224, China; 3Yunnan Key Laboratory for Fungal Diversity and Green Development, Kunming Institute of Botany, Chinese Academy of Sciences, Kunming 650201, China

**Keywords:** Asia, macrofungi, molecular systematics, taxonomy, Yunnan Province

## Abstract

Four new fungi of the family Peniophoraceae, viz., *Peniophora roseoalba*, *P. yunnanensis*, *Vararia daweishanensis*, and *V. fragilis* are herein proposed, based on a combination of morphological features and molecular evidence. *Peniophora roseoalba* is characterized by resupinate, membranaceous basidiomata with a rose pink to pale pinkish grey hymenophore, a monomitic hyphal system with clamped generative hyphae, the presence of cystidia, and ellipsoid basidiospores. However, *P. yunnanensis* differs in being tuberculate, with a pale cream to cream hymenial surface, small lamprocystidia (18–29 × 4.5–7 µm), and subcylindrical basidiospores. *Vararia daweishanensis* is characterized by resupinate, membranous basidiomata with a pale yellowish hymenial surface, a dimitic hyphal system with clamped generative hyphae, strongly dextrinoid dichohyphae, and allantoid basidiospores; *V. fragilis* is characterized by resupinate, brittle basidiomata, with a buff to ochraceous hymenial surface and small ellipsoid basidiospores measuring 3.5–5.5 × 2.5–3.5 µm. Sequences of the ITS and nLSU rRNA markers of the studied samples were generated, and phylogenetic analyses were performed with the maximum likelihood, maximum parsimony, and Bayesian inference methods. The nLSU analysis revealed that the four new species can be clustered into the family Peniophoraceae (Russulales), in the genera *Peniophora* and *Vararia*. Further studies based on the ITS dataset showed that four fungi of the family Peniophoraceae were new to science.

## 1. Introduction

The family Peniophoraceae (Russulales) is a large and rather heterogeneous family with seven genera accepted; two genera, *Peniophora* Cooke and *Vararia* P. Karst., have the highest number of taxa in this family, in which they play fundamental ecological roles to drive carbon cycling in forest soils, acting as decomposers [1,2].

*Peniophora*, typified by *P. quercina* (Pers.) Cooke, is characterized by resupinate, membranaceous to ceraceous basidiomata, with smooth to tuberculate hymenophores having a grey, violaceous, orange, red, or brown hymenial surface, a monomitic hyphal system with clamped generative hyphae; dendrohyphidia, lamprocystidia, and gloeocystidia are present or absent; the basidiospores are ellipsoid, cylindrical to allantoid, smooth, thin-walled, acyanophilous, and without reaction with Melzer [3]. Based on the MycoBank database (http://www.MycoBank.org, accessed on 13 October 2022) and the Index Fungorum (http://www.indexfungorum.org, accessed on 13 October 2022), the genus *Peniophora* has 637 specific and registered names, but the actual number of species has reached 191 [4,5,6,7,8,9,10,11,12,13,14,15,16,17,18]. *Vararia* is typified by *V. investiens* (Schwein.) P. Karst. This genus is characterized by resupinate basidiomata, a dimitic hyphal system with clamped or simple-septate generative hyphae and often, dextrinoid dichohyphae, the presence of gloeocystidia, and variously shaped smooth basidiospores with or without an amyloid reaction [19,20,21]. The MycoBank database (http://www.MycoBank.org, accessed on 13 October 2022) and Index Fungorum (http://www.indexfungorum.org, accessed on 13 October 2022) have registered 96 specific and infraspecific names in *Vararia*, but the actual number of the species has reached 72, and they occur mainly in the tropical and subtropical areas of the world [22,23,24,25,26,27,28,29,30,31]. However, *Vararia* is still poorly studied in China [32], from whence eight species, namely, *V. amphithallica* Boidin, Lanq. & Gilles, *V. bispora* S.L. Liu & S.H. He, *V. breviphysa* Boidin & Lanq., *V. cinnamomea* Boidin, Lanq. & Gilles, *V. investiens* (Schwein.) P. Karst., *V. montana* S.L. Liu & S.H. He, *V. racemosa* (Burt.) D.P. Rogers & H.S. Jacks., and *V. sphaericospora* Gilb., have been reported in this country [32,33,34].

These pioneering research studies into the Peniophoraceae family were just the prelude to the molecular systematics period. The phylogenetic diversity displayed by corticioid fungal species, based on 5.8S and 28S nuclear rDNA, revealed that the taxa of Peniophoraceae are nested in the russuloid clade, which holds a considerable share of the phylogenetic framework, and include the genera of *Peniophora* and *Vararia* [35]. The phylogenetic research about the major clades of mushroom-forming fungi (Homobasidiomycetes) indicated that the largest resupinate forms divided into the polyporoid, russuloid, and hymenochaetoid clades, in which *Peniophora* grouped with *Asterostroma* Massee and *Scytinostroma* Donk [36]. Molecular phylogenetic analyses of nrITS and nrLSU sequences revealed affinities among families with the Peniophorales in the Russulales, in which the presence of distinctive hyphal elements, which are homologous to the defining features of Peniophorales, was consistent with the phylogenetic evidence, and the Varariaceae were grouped closely with the Peniophoraceae [37].

During the investigations into wood-inhabiting fungi in Yunnan Province, China, four new taxa of Peniophoraceae were found that could not be assigned to any described species. Herein, we present the morphological and molecular phylogenetic evidence that supports the recognition of these four new species within the *Peniophora* and *Vararia*, based on the internal transcribed spacer (ITS) regions and the large subunit nuclear ribosomal RNA gene (nLSU) sequences.

## 2. Materials and Methods

### 2.1. Morphology

Fresh fruiting bodies of the fungi were collected from Chuxiong, Honghe, Puer, and Wenshan of Yunnan Province, in China. The specimens were dried in an electric food dehydrator at 40 °C, then sealed and stored in an envelope bag and deposited in the herbarium of the Southwest Forestry University (SWFC), Kunming, Yunnan Province, China. The macromorphological descriptions are based on field notes and photos captured in the field and laboratory. The color terminology follows the example set by Petersen [38,39,40]. Micromorphological data were obtained from the dried specimens when observed under a light microscope, following the method used by Dai [41]. The following abbreviations were used: KOH = 5% potassium hydroxide water solution, CB = Cotton Blue, CB− = acyanophilous, CB+ = cyanophilous, IKI = Melzer’s reagent, IKI− = both inamyloid and indextrinoid, L = mean spore length (arithmetic average for all spores), W = mean spore width (arithmetic average for all spores), Q = variation in the *L*/*W* ratios between the specimens studied, and *n* = a/b (number of spores (a) measured from a given number (b) of specimens).

### 2.2. Molecular Phylogeny

The CTAB rapid plant genome extraction kit-DN14 (Aidlab Biotechnologies Co., Ltd., Beijing, China) was used to obtain the genomic DNA from the dried specimens using the manufacturer’s instructions, following a previous study [42]. The nuclear ribosomal ITS region was amplified with the primers ITS5 and ITS4 [43]. The nuclear nLSU region was amplified with the primer pair, LR0R and LR7 (http://lutzonilab.org/nuclear-ribosomal-dna/; accessed on 13 October 2022). The PCR procedure for ITS was as follows: initial denaturation at 95 °C for 3 min, followed by 35 cycles at 94 °C for 40 s, 58 °C for 45 s, and 72 °C for 1 min, with a final extension of 72 °C for 10 min. The PCR procedure for nLSU was as follows: initial denaturation at 94 °C for 1 min, followed by 35 cycles at 94 °C for 30 s, 48 °C for 1 min, and 72 °C for 1.5 min, with a final extension of 72 °C for 10 min. The PCR products were purified and sequenced at Kunming Tsingke Biological Technology Company, Limited (Yunnan Province, China). All of the newly generated sequences were deposited in the NCBI GenBank (https://www.ncbi.nlm.nih.gov/genbank/; accessed on 13 October 2022) (Table 1).

The sequencer, 4.6 (GeneCodes, Ann Arbor, MI, USA), was used to assemble and edit the generated sequence reads. The sequences were aligned in MAFFT 7 (https://mafft.cbrc.jp/alignment/server/; accessed on 13 October 2022), using the “G-INS-i” strategy for the ITS and nLSU dataset, manually adjusted in BioEdit [44]. The sequences of *Sistotrema brinkmannii* (Bres.) J. Erikss. and *S. coronilla* (Höhn.) Donk ex. D.P. Rogers, obtained from GenBank, were selected as an outgroup for the phylogenetic analysis of the nLSU phylogenetic tree (Figure 1) [45]; the sequences of *Dichostereum durum* (Bourdot & Galzin) Pilát and *D. effuscatum* (Cooke & Ellis) Boidin & Lanq. were selected as an outgroup for phylogenetic analysis of ITS phylogenetic tree (Figure 2) [45]; the sequences of *P. incarnata* (Pers.) P. Karst. and *P. nuda* (Fr.) Bres. were selected as an outgroup in the ITS analysis (Figure 3), following the method of a previous study [38].

Maximum parsimony (MP), maximum likelihood (ML), and Bayesian inference (BI) analyses were applied to the three combined datasets, following the technique used in a previous study [42], and the tree construction procedure was performed in PAUP*, version 4.0b10 [46]. All the characters were equally weighted, and gaps were treated as missing data. Trees were inferred using the heuristic search option, with TBR branch swapping and 1000 random sequence additions. The max trees were set to 5000, branches of zero length were collapsed, and all parsimonious trees were saved. Clade robustness was assessed using a bootstrap (BT) analysis with 1000 replicates [47]. The descriptive tree statistics were tree length (TL), consistency index (CI), retention index (RI), rescaled consistency index (RC), and homoplasy index (HI); these were calculated for each maximum parsimonious tree generated. The multiple sequence alignment was also analyzed using the maximum likelihood (ML) in RAxML-HPC2, through the Cipres Science Gateway (www.phylo.org; accessed on 13 October 2022) [48]. Branch support (BS) for the ML analysis was determined by 1000 bootstrap replicates.

**Table 1 jof-08-01227-t001:** List of species, specimens, and GenBank accession numbers of the sequences used in this study.

Species Name	Specimen No.	GenBank Accession No.		References	Country
		ITS	nLSU		
*Amylostereum areolatum*	NH 8041	AF506405	AF506405	[45]	Sweden
*A. chailletii*	NH 8031	AF506406	AF506406	[45]	Sweden
*A. laevigatum*	NH 12863	AF506407	AF506407	[45]	Sweden
*Asterostroma bambusicola*	He 4132		KY263871	this publication	Thailand
*A. cervicolor*	He 2314		KY263869	this publication	China
*A. laxum*	EL 33-99	AF506410	AF506410	[45]	Sweden
*A. vararioides*	He 4140		KY263870	this publication	Thailand
*Auriscalpium vulgare*	EL 33-95	AF506375	AF506375	[45]	Sweden
*Baltazaria galactina*	CBS: 752.86		MH873721	[49]	France
*B. neogalactina*	CBS: 755.86	MH873724	MH873724	[49]	French
*B. occidentalis*	AFTOL-ID		DQ234539	[50]	Canada
*B. podocarpi*	Dai 9261		KJ583221	[51]	China
*Dentipratulum bialoviesense*	GG 1645	AF506389	AF506389	[45]	Sweden
*Dichostereum durum*	FG 1985	AF506429	AF506429	[45]	Sweden
*D. effuscatum*	GG 930915	AF506390	AF506390	[45]	Sweden
*Gloeocystidiellum bisporum*	KHL 11135	AY048877	AY048877	[45]	Sweden
*G. clavuligerum*	FCUG 2159	AF310088	AF310088	[52]	Spain
*G. purpureum*	Wu 9310-45	AF441338	AF441338	[45]	China
*Gloeocystidiopsis flammea*	CBS: 324.66	AF506437	AF506437	[45]	C. African Rep.
*Gloeodontia columbiensis*	NH 11118	AF506444	AF506444	[45]	Spain
*G. discolor*	KHL 10099	AF506445	AF506445	[45]	USA
*G. eriobotryae*	Dai 12080		JQ349103	[53]	China
*G. pyramidata*	LR 15502	AF506446	AF506446	[45]	Colombia
*G. subasperispora*	KHL 8695	AF506404	AF506404	[45]	Norway
*G. yunnanensis*	SWFC 00010504		MN908254	[54]	China
*Gloeopeniophorella* *convolvens*	KHL 10103	AF506435	AF506435	[45]	USA
*Gloiothele lactescens*	EL 8-98	AF506453	AF506453	[45]	Sweden
*G. lamellosa*	KHL 11031	AF506454	AF506454	[45]	Venezuela
*Heterobasidion annosum*	06129/6		KJ583225	[51]	Russia
*H. parviporum*	04121/3		KJ583226	[51]	Finland
*Lachnocladium schweinfurthianum*	KM 49740		MH260051	[38]	Cameroon
*Lactarius leonis*	SJ 91016	AF506411	AF506411	[45]	Sweden
*Lentinellus cochleatus*	KGN 960928	AF506417	AF506417	[45]	Sweden
*L. ursinus*	EL 73-97	AF506419	AF506419	[45]	USA
*L. vulpinus*	KGN 980825 (GB)	AF347097	AF347097	[45]	Sweden
*Megalocystidium luridum*	KHL 8635	AF506422	AF506422	[45]	Norway
*Michenera artocreas*	GHL-2016-Oct		MH204692	[55]	USA
*M. incrustata*	He 2630		MH142907	[55]	China
*Peniophora albobadia*	CBS: 329.66	MH858809	MH858809	[49]	France
*P. bicornis*	He 3609	MK588763	MK588763	[39]	China
*P. bicornis*	He 4767	MK588764	MK588764	[39]	China
*P. borbonica*	He 4597	MK588766	MK588766	[39]	China
*P. borbonica*	He 4606	MK588765	MK588765	[39]	China
*P. cinerea*	B 1020	MN475151	MN475151	[39]	USA
*P. crassitunicata*	CBS: 663.91	MH862292	MH862292	[49]	France
*P. duplex*	B 1022	MN475153	MN475153	[39]	USA
*P. erikssonii*	CBS: 287.58	MH857788	MH857788	[39]	France
*P.erikssonii*	Cui 11871	MK588771	MK588811	[39]	China
*P. exima*	B 1011	MN475155	MN475155	[39]	USA
*P. exima*	T-523	MK588772	MK588772	[39]	USA
*P. fasticata*	CBS: 942.96	MH862624	MH862624	[39]	Ethiopia
*P. fissilis*	CBS: 681.91	MH862298	MH862298	[39]	France
*P. fissilis*	CBS: 684.91	MH862299	MH862299	[39]	Netherlands
*P. gabonensis*	CBS: 673.91	MH862293	MH862293	[39]	Gabon
*P. gilbertsonii*	CBS: 357.95	MH862528	MH862528	[39]	USA
*P. gilbertsonii*	CBS: 360.95	MH862530	MH862530	[39]	USA
*P. halimi*	CBS: 862.84	MH861843	MH861843	[39]	France
*P. incarnata*	NH 10271	AF506425	AF506425	[45]	Denmark
*P. incarnata*	CBS: 430.72	MH860518	MH872230	[39]	Netherlands
*P. junipericola*	He 2462	MK588773	MK588773	this publication	China
*P. laeta*	CBS: 256.56	MH857617	MH857617	[39]	France
*P. laeta*	CBS: 255.56	MH857616	MH857616	[39]	France
*P. laxitexta*	LGMF 1159	JX559580		[39]	Brazil
*P. laxitexta*	BAFC 3309	FJ882040		[39]	Argentina
*P. laxitexta*	BAFC: 4687	MN518328		[39]	Argentina
*P. lilacea*	CBS: 337.66	MH858813	MH858813	[39]	Armenia
*P. lycii*	CBS: 264.56	MH857624	MH857624	[39]	France
*P. lycii*	CBS: 261.56	MH857621	MH857621	[39]	France
*P. malaiensis*	He 4870	MK588775	MK588775	[39]	China
*P. manshurica*	He 2956	MK588776	MK588776	[39]	China
*P. manshurica*	He 3729	MK588777	MK588777	[39]	China
*P. meridionalis*	CBS: 289.58	MH857789	MH857789	[49]	France
*P. molesta*	CBS: 678.91	MH862296	MH862296	[39]	Cote d’Ivoire
*P. molesta*	CBS: 676.91	MH862294	MH862294	[39]	Gabon
*P. molesta*	CBS: 677.91	MH862295	MH862295	[39]	Gabon
*P. monticola*	CBS: 649.91	MH862289	MH862289	[39]	France
*P. nuda*	AFTOL-ID 660	DQ411533		[39]	USA
*P. nuda*	LZ15-07	MT859929	MT859929	this publication	China
*P. ovalispora*	CBS: 653.91		MH873971	[39]	Netherlands
*P. ovalispora*	CBS: 653.91	MH862290	MH862290	[39]	Netherlands
*P. parvocystidiata*	CBS: 716.91	MH862305	MH862305	[39]	France
*P. parvocystidiata*	CBS: 717.91	MH862306	MH862306	[39]	France
*P. piceae*	B 1010	MN475158	MN475158	this publication	USA
*P. pilatiana*	CBS: 269.56	MH857627	MH857627	[39]	France
*P. pilatiana*	CBS: 265.56	MH857625	MH857625	[39]	France
*P. pilatiana*	CBS: 266.56	MH857626	MH857626	[39]	France
*P. pini*	CBS: 273.56	MH857631	MH857631	[39]	France
*P. pini*	CBS: 270.56	MH857628	MH857628	[39]	France
*P. pithya*	CBS: 275.56	MH857633	MH857633	[49]	France
*P. polygonia*	He 3668	MH669233		[56]	China
*P. polygonia*	CBS: 404.50	MH856684	MH856684	[39]	France
*P. proxima*	CBS: 406.50	MH856686	MH856686	[39]	France
*P. proxima*	CBS: 405.50	MH856685	MH856685	[39]	France
*P. pseudonuda*	FCUG 2384	GU322866		this publication	Sweden
*P. pseudonuda*	FCUG 2390	GU322865		this publication	Sweden
*P. pseudopini*	B 1024	MN475163	MN475163	this publication	USA
*P. pseudoversicolor*	CBS: 125881	MH864303	MH864303	[39]	France
*P. quercina*	CBS: 407.50	MH856687	MH868204	[39]	France
*P. quercina*	CBS: 408.50	MH856688	MH856688	[39]	France
*P. quercina*	CBS: 409.50	MH856689	MH856689	[39]	France
*P. reidii*	CBS: 397.83	MH861616	MH861616	[39]	France
*P. rosealba*	CLZhao 3513	ON786559	OP380690	present study	China
*P. rosealba*	CLZhao 9401 *	ON786560		present study	China
*P. rufa*	B 1014	MN475165	MN475165	this publication	USA
*P. rufa*	CBS: 351.59	MH857891	MH869432	[39]	Canada
*P. rufomarginata*	CBS: 281.56	MH857639	MH857639	[39]	France
*P. rufomarginata*	CBS: 282.56	MH857640	MH857640	[39]	France
*P. septentrionalis*	CBS: 294.58	MH857791	MH857791	[39]	Canada
*P. simulans*	CBS: 875.84	MH861850	MH861850	[39]	France
*P. simulans*	CBS: 874.84	MH861849	MH861849	[39]	France
*P. subsalmonea*	CBS: 697.91	MH862303	MH862303	[39]	Netherlands
*P. subsalmonea*	CBS: 696.91	MH862302	MH862302	[39]	Netherlands
*P. taiwanensis*	Wu 9209-14	MK588794	MK588794	[39]	China
*P. tamaricicola*	CBS: 438.62	MH858203	MH858203	[39]	Morocco
*P. tamaricicola*	CBS: 439.62	MH858204	MH858204	[39]	Morocco
*P. tamaricicola*	CBS: 441.62	MH858205	MH858205	[39]	Morocco
*P. versicolor*	CBS: 358.61	MH858082	MH858082	[39]	Morocco
*P. violaceolivida*	CBS: 348.52	MH857077	MH857077	[39]	France
*P. yunnanensis*	CLZhao 3978	OP380617	OP380689	present study	China
*P. yunnanensis*	CLZhao 7347 *	OP380616		present study	China
*P. yunnanensis*	CLZhao 8135	OP380615		present study	China
*Russula violacea*	SJ 93009	AF506465	AF506465	[45]	Sweden
*Scytinostroma portentosum*	EL 11-99	AF506470	AF506470	[45]	Sweden
*Sistotrema brinkmannii*	NH 11412	AF506473	AF506473	[45]	Turkey
*S. coronilla*	NH 7598	AF506475	AF506475	[45]	Canada
*Stereum hirsutum*	NH 7960	AF506479	AF506479	[45]	Romania
*Vararia abortiphysa*	CBS: 632.81	MH861387	MH861387	[49]	Gabon
*V. ambigua*	CBS: 634.81	MH861388	MH873137	[49]	France
*V. amphithallica*	CBS: 687.81	MH861431	MH861431	[49]	France
*V. aurantiaca*	CBS: 642.81	MH861394	MH861394	[49]	Gabon
*V. aurantiaca*	CBS: 641.81	MH861393	MH861393	[49]	France
*V. breviphysa*	CBS: 644.81	MH861396	MH861396	[49]	Gabon
*V. calami*	CBS: 646.81	MH861398	MH861398	[49]	France
*V. calami*	CBS: 648.81	MH861399	MH861399	[49]	France
*V. callichroa*	CBS: 744.91	MH874000	MH874000	[49]	France
*V. cinnamomea*	CBS: 642.84	MH873488	MH873488	[49]	Madagascar
*V. cinnamomea*	CBS: 641.84	MH861794	MH861794	[49]	Madagascar
*V. cremea*	CBS: 651.81	MH873147	MH873147	[49]	France
*V. daweishanensis*	CLZhao 17911	OP380613	OP615103	present study	China
*V. daweishanensis*	CLZhao 17936 *	OP380614	OP380688	present study	China
*V. dussii*	CBS: 655.81	MH861405	MH861405	[49]	France
*V. dussii*	CBS: 652.81	MH873148	MH873148	[49]	France
*V. ellipsospora*	HHB-19503	MW740328	MW740328	this publication	New Zealand
*V. fragilis*	CLZhao 2628	OP380611		present study	China
*V. fragilis*	CLZhao 16475 ***	OP380612	OP380687	present study	China
*V. fusispora*	PDD: 119539	OL709443	OL709443	this publication	New Zealand
*V. gallica*	CBS: 234.91	MH862250		[49]	Canada
*V. gallica*	CBS: 656.81	MH861406	MH873152	[49]	France
*V. gillesii*	CBS: 660.81	MH873153	MH873153	[49]	Cote d’Ivoire
*V. gomezii*	CBS: 661.81	MH873154	MH873154	[49]	French
*V. gracilispora*	CBS: 664.81	MH861412	MH861412	[49]	Gabon
*V. gracilispora*	CBS: 663.81	MH861411		[49]	Gabon
*V. insolita*	CBS: 668.81	MH861413	MH861413	[49]	France
*V. intricata*	CBS: 673.81	MH861418	MH861418	[49]	France
*V. investiens*	FP-151122	MH971976	MH971977	[56]	USA
*V. malaysiana*	CBS: 644.84	MH873490	MH873490	[49]	Singapore
*V. minispora*	CBS: 682.81	MH861426	MH861426	[49]	France
*V. ochroleuca*	CBS: 465.61	MH858109	MH858109	[49]	France
*V. ochroleuca*	JS 24400	AF506485	AF506485	[45]	Norway
*V. parmastoi*	CBS: 879.84	MH861852	MH861852	[49]	Uzbekistan
*V. perplexa*	CBS: 695.81	MH861438	MH861438	[49]	France
*V. pectinata*	CBS: 685.81	MH861429		[49]	Cote d’Ivoire
*V. pirispora*	CBS: 720.86	MH862016	MH862016	[49]	France
*V. rhombospora*	CBS: 743.81	MH861470	MH861470	[49]	France
*V. rosulenta*	CBS: 743.86	MH862028		[49]	France
*V. rugosispora*	CBS: 697.81	MH861440	MH861440	[49]	Gabon
*V. sigmatospora*	CBS: 748.91	MH874001	MH874001	[49]	Netherlands
*V. sphaericospora*	CBS: 700.81	MH873185	MH873185	[49]	Gabon
*V. sphaericospora*	CBS: 703.81	MH861446	MH861446	[49]	Gabon
*V. trinidadensis*	CBS: 651.84	MH861803	MH861803	[49]	Madagascar
*V. trinidadensis*	CBS: 650.84	MH873495	MH873495	[49]	Madagascar
*V. tropica*	CBS: 704.81	MH861447	MH873189	[49]	France
*V. vassilievae*	UC2022892	KP814203	KP814203	this publication	USA
*V. verrucosa*	CBS 706.81	MH861449	MH861449	[49]	France
*Vesiculomyces citrinus*	EL 53-97	AF506486	AF506486	[45]	Sweden

* indicates the holotype.

MrModeltest 2.3 [57] was used to determine the best-fit evolution model for each dataset, using Bayesian inference (BI), which was performed using MrBayes 3.2.7a, with a GTR+I+G model of the DNA substitution and a gamma distribution rate variation across the sites [58]. Four Markov chains were run for 2 runs, beginning from random starting trees for 0.9 million generations for nLSU (Figure 1), for 1.5 million generations for ITS (Figure 2) with trees, and 1 million generations for ITS (Figure 3) with trees, with the parameters sampled every 1000 generations. The first one-quarter of all generations were discarded as the burn-in. The majority rule consensus tree of all the remaining trees was calculated. Branches were considered significantly supported if they received a maximum likelihood bootstrap value (BS) > 70%, a maximum parsimony bootstrap value (BT) > 70%, or Bayesian posterior probabilities (BPP) > 0.95.

## 3. Results

### 3.1. Molecular Phylogeny

The nLSU dataset (Figure 1) included sequences from 55 fungal specimens, representing 55 species. The dataset had an aligned length of 1415 characters, of which 923 characters are constant, 152 are variable and parsimony-uninformative, and 340 are parsimony-informative. The maximum parsimony analysis yielded one equally parsimonious tree (TL = 860, CI = 0.3233, HI = 0.6767, RI = 0.6123, RC = 0.1979). The best model for the ITS+nLSU dataset, which was estimated and applied in the Bayesian analysis, was GTR+I+G (lset nst = 6, rates = invgamma; prset statefreqpr = dirichlet (1,1,1,1). The Bayesian analysis and ML analysis resulted in a similar topology to the MP analysis, with an average standard deviation of split frequencies = 0.009575 (BI); the effective sample size (ESS) across the two runs is the double of the average ESS (avg ESS) = 200.5. The phylogeny (Figure 1), based on the combined nLSU sequences, includes six families within the order of Russulales, which indicated that nine genera, comprising *Asterostroma* Massee, *Baltazaria* Leal-Dutra, Dentinger & G.W. Griff., *Gloiothele* Bres., *Lachnocladium* Lév., *Michenera* Berk. & M.A. Curtis, *Peniophora*, *Scytinostroma* Donk, *Vararia*, and *Vesiculomyces* E. Hagstr. could be incorporated into the Peniophoraceae family. Our current four new species can be clustered into the genera of *Peniophora* and *Vararia*, respectively.

The ITS-alone dataset of the genus *Peniophora* (Figure 2) included the sequences from 83 fungal specimens, representing 52 species. The dataset had an aligned length of 607 characters, of which 353 characters were constant, while 64 were variable and parsimony-uninformative, and 190 were parsimony-informative. The maximum parsimony analysis yielded 12 equally parsimonious trees (TL = 1681, CI = 0.3111, HI = 0.6889, RI = 0.4496, RC = 0.1399). The best model for the ITS dataset that was estimated and applied in the Bayesian analysis was GTR+I+G (lset nst = 6, rates = invgamma; prset statefreqpr = dirichlet (1,1,1,1). Bayesian analysis and ML analysis resulted in a similar topology to MP analysis, with an average standard deviation of split frequencies = 0.009599 (BI). The phylogenetic tree indicated that *P. roseoalba* can be grouped with two close taxa, *P*. *versicolor* and *P. ovalispora*, whereas *P. yunnanensis* can be grouped with a clade comprising *P. lycii* and *P*. *violaceolivida*.

The ITS-only dataset of the genus *Vararia* (Figure 3) included sequences from 63 fungal specimens, representing 39 species. The dataset had an aligned length of 1128 characters, of which 511 characters were constant, 133 were variable and parsimony uninformative, and 484 were parsimony informative. Maximum parsimony analysis yielded 6 equally parsimonious trees (TL = 4589, CI = 0.2805, HI = 0.7195, RI = 0.4174, and RC = 0.1171). The best model for the ITS dataset estimated and applied in the Bayesian analysis was GTR+I+G. The Bayesian and ML analyses resulted in a similar topology to that of the MP analysis with split frequencies = 0.0096082 (BI). The phylogram inferred from the ITS sequences (Figure 3) revealed that *Vararia daweishanensis* could be grouped with four close taxa: *V*. *gomezii*, *V*. *rhombospora*, *V*. *sigmatospora,* and *V*. *trinidadensis*, whereas the other species of *V*. *fragilis* could be grouped with a clade comprising *V. ambigua* and *V. ellipsospora*, with a low level of support.

### 3.2. Taxonomy

***Peniophora roseoalba*** L. Zou & C.L. Zhao, sp. nov. (Figure 4 and Figure 5).

MycoBank no.: 845758.

**Holotype—**China, Yunnan Province, Puer, Jingdong county, the Forest of Pineapple, 24°37′ N, 100°45′ E, altitude 2083 m asl., on the fallen branch of an angiosperm, 4 January 2019, CLZhao 9401 (SWFC).

**Etymology—*Roseoalba*** (Lat.): referring to the rose to pale pinkish grey color of the hymenial surface of the specimens.

**Fruiting body—**Basidiomata are annual, resupinate, membranaceous, without odor and taste when fresh, up to 90 mm long, 20 mm wide, 70–100 µm thick. The hymenial surface is smooth, occasionally cracked, and rose to pale pinkish grey. The sterile margin is indistinct and is rose to pinkish grey.

**Hyphal system—**Monomitic, generative hyphae with clamp connections, colorless, thin- to thick-walled, moderately branched, 1.5–4.5 µm in diameter, CB−, IKI−; tissues unchanged in KOH.

**Hymenium—**The cystidia are of two types: (1) Gloeocystidia is subcylindrical to conical, smooth, colorless, thin-walled, 31.5–40.5 × 6.5–7.5 µm; (2) Lamprocystidia is abundant in the hymenium, and is conical, thick-walled, encrusted apical part, colorless, 33–42.5 × 7–10.5 µm. The Basidia are subclavate to subcylindrical, slightly constricted in the middle, with four sterigmata and a basal clamp connection, sized 24–39.5 × 3.5–5.5 µm.

**Basidiospores—**Basidiospores are ellipsoid, colorless, thin-walled, smooth, IKI−, CB−, 4–6.5 × 3–5 µm, L = 5.19 µm, W = 3.8 µm, Q = 1.26–1.48 (*n* = 60/2).

**Additional specimen examined—**China, Yunnan Province, Puer, Jingdong County, Wuliangshan National Nature Reserve, 23°57′ N, 100°22′ E, altitude 3376 m asl., found on the fallen branch of an angiosperm, 2 October 2017, CLZhao 3513 (SWFC).

***Peniophora yunnanensis*** L. Zou & C.L. Zhao sp. nov. (Figure 6 and Figure 7).

MycoBank no.: 845760.

**Holotype—**China, Yunnan Province, Chuxiong, Zixishan Forestry Park, 25°01′ N, 101°24′ E., altitude 2356 m asl., on an angiosperm stump, 2 July 2018, code: CLZhao 7347 (SWFC).

**Etymology—*Yunnanensis*** (Lat.): referring to the geographic provenance (Yunnan Province) of the specimens.

**Fruiting body—**Basidiomata are annual, resupinate, and coriaceous, without odor and taste when fresh, up to 100 mm long, 25 mm wide, and 70–100 µm thick. The hymenial surface is tuberculate and is pale cream to cream. The sterile margin is indistinct and slightly cream-colored.

**Hyphal system—**Monomitic, generative hyphae with clamp connections, colorless, thin- to thick-walled, moderately branched, 2.5–3.5 µm in diameter, IKI−, CB−; tissues are unchanged in the KOH; the subiculum generative hyphae are dense, with a subparallel arrangement; the subhymenium is composed of strongly agglutinated vertical hyphae.

**Hymenium—**The cystidia are of two types: (1) Gloeocystidia, which are different in shape, conical, clavate to fusiform, and subglobose, usually containing refractive materials; they are colorless, smooth, thin-walled, and 12.5–58 × 5.5–15.5 µm; (2) Lamprocystidia are abundant in the hymenium, the conical, thick-walled, encrusted apical part, colorless, and 18–29 × 4.5–7 µm. The basidia subclavate changes to subcylindrical, being slightly constricted in the middle to somewhat constricted, with four sterigmata and a basal clamp connection, 22.5–39.5 × 4.5–8 µm.

**Basidiospores—**The basidiospores are subcylindrical, colorless, thin-walled, and smooth, with oil drops occasionally found inside, IKI−, CB−, (5–) 5.5–10 (–11) × 3–5.5 µm, L = 7.72 µm, W = 4.44 µm, Q = 1.61–1.88 (*n* = 90/3).

**Additional specimens examined (paratypes)—**China, Yunnan Province, Puer, Jingdong County, Taizhong Town, Ailaoshan, 24°23′ N, 120°53′ E, altitude 3166 m asl.; found on the fallen branch of an angiosperm, 4 October 2017, CLZhao 3978 (SWFC). Zhenyuan County, Ailaoshan, Jinshan Original Forestry, 24°00′ N, 101°10′ E; altitude 2300 m asl., and found on the fallen branch of an angiosperm, 21 August 2018, CLZhao 8135 (SWFC).

***Vararia daweishanensis*** L. Zou & C.L. Zhao, sp. nov. (Figure 8 and Figure 9).

MycoBank no.: 845761.

**Holotype—**China, Yunnan Province, Honghe, Pinbian County, Daweishan National Forestry Park, 22°53′ N, 103°35′ E, altitude 1670 m asl., found on a fallen angiosperm branch, 1 August 2019, CLZhao 17936 (SWFC).

**Etymology—*daweishanensis*** (Lat.): referring to the provenance (Daweishan) of the specimens.

**Fruiting body—**Basidiomata are annual, resupinate, membranous, soft, and adnate, up to 80 mm long, 16 mm wide, and 70–150 µm thick. The hymenial surface is smooth and pale yellowish. The sterile margin is distinct, narrow, whitish, and attached.

**Hyphal system—**Dimitic, generative hyphae with clamp connections, colorless, thin- to thick-walled, occasionally branched, interwoven, 2–4 µm in diameter, IKI−, CB+, tissues are unchanged in KOH; dichohyphae in subhymenium abundant, yellowish, capillary, distinctly thick-walled; dichotomously to irregularly branched, with the main branches up to 4 μm in diameter and with acute tips, moderately dextrinoid when in Melzer’s reagent; more frequently branched with more narrow and shorter branches in the hymenium, with slightly curved tips and a stronger dextrinoid reaction.

**Hymenium—**The gloeocystidia are empty or are filled with a refractive oil-like matter; they are also subcylindrical. The hymenium is elliptical to ovoid, smooth, colorless, thin-walled, and 9–23 × 7–10.5 µm. The basidia are subcylindrical, with four sterigmata and a basal clamp connection, 26–46 × 5–8 µm.

**Basidiospores—**The basidiospores are allantoid, colorless, thin-walled, and smooth, with oil droplets inside, IKI−, CB−, (8.5–) 9–13 (–14) × 3.5–5 µm, L = 10.57 µm, W = 4.23 µm, Q = 2.44–2.55 (*n* = 60/2).

**Additional specimens examined (paratypes)—**China, Yunnan Province, Honghe, Pinbian County, Daweishan National Forestry Park, 22°53′ N, 103°35′ E, altitude 1670 m asl., found on a fallen angiosperm branch, 1 August 2019, CLZhao 17911 (SWFC).

***Vararia fragilis*** L. Zou & C.L. Zhao, sp. nov. (Figure 10 and Figure 11).

MycoBank no.: 845763.

**Holotype—**China, Yunnan Province, Wenshan, Wenshan National Nature Reserve. GPS coordinates: found at 23°22′ N, 104°43′ E, altitude 1500 m asl., found on the fallen branch of an angiosperm, 26 July 2019, CLZhao 16475 (SWFC).

**Etymology—*fragilis*** (Lat.): referring to the fragile basidiomata.

**Fruiting body—**Basidiomata are annual, resupinate, adnate, thin, membranous, and fragile, without odor and taste when fresh, up to 85 mm long, 40 mm wide, and 30–100 µm thick. The hymenial surface is smooth, buff when fresh, buff to ochraceous on drying and cracking. The sterile margin is indistinct, attached, and is cream to buff.

**Hyphal system—**Dimitic, generative hyphae, bearing simple septa, colorless, thin- to thick-walled, occasionally branched, interwoven, 1.5–3.5 µm in diameter, IKI−, CB+, and with tissues unchanged in KOH; the dichohyphae in the subhymenium are abundant, predominantly yellowish, capillary, distinctly thick-walled, and dichotomously to irregularly branched, with the main branches up to 2 μm in diameter and with acute tips; moderately dextrinoid in Melzer’s reagent; more frequently branched, with more narrow and shorter branches in the subiculum, with slightly curved tips and a stronger dextrinoid reaction.

**Hymenium—**The gloeocystidia are of two types: (1) elliptical to ovoid, 5.8–16 × 3.5–7 µm; (2) subulate, usually with a constriction at the tip, smooth, colorless, thin-walled, 16.5–27 × 4–7 µm. Basidia subcylindrical, with four sterigmata and a basal simple-septa connection, 13–23.5 × 3–4.5 µm.

**Basidiospores—**The basidiospores are broad from ellipsoid to ellipsoid, colorless, thin-walled, smooth, IKI−, CB−, 3.5–5.5 (–6) × 2.5–3.5 µm, L = 4.78 µm, W = 3.12 µm, Q = 1.48–1.56 (*n* = 60/2).

**Additional specimen examined (paratype)—**China, Yunnan Province, Yuxi, Xiping County, Mopanshan National Forestry Park, 24°07′ N, 101°98′ E, altitude 2614 m asl., on the fallen branch of an angiosperm, 20 August 2017, CLZhao 2628 (SWFC).

## 4. Discussion

Four genera, *Gloiothele*, *Peniophora*, *Scytinostroman*, and *Vararia* have been grouped together and clustered within the family Peniophoraceae, as inferred from a dataset with 178 terminal taxa [37]. In the present study, based on the nLSU data (Figure 1), four new species were classified in the family Peniophoraceae and were then classified within the genera of *Peniophora* and *Vararia*.

Based on the ITS phylogenetic analysis (Figure 2), two new taxa have been grouped within the genus *Peniophora*, named *P*. *roseoalba* and *P. yunnanensis*, in which *P. roseoalba* is grouped with two close taxa, *P. versicolor* and *P. ovalispora*; *P. yunnanensis* was grouped with a clade comprising *P. lycii* and *P. violaceolivida*. However, morphologically, *Peniophora versicolor* differs from *P. roseoalba* by its dark brown to reddish brown or ochraceous hymenophore, smaller lamprocystidia (10–20 × 8–10 µm), and larger basidiospores (9–11 × 4.5–6 µm) [59]. *P. ovalispora* is separated from *P. roseoalba* by having a cream-colored to salmon or brownish hymenophore, with a pruinose margin [13,60]. *Peniophora lycii* is separated from *P. yunnanensis* by its even, greyish lilac to bluish violaceous hymenial surface, the presence of the dendrohyphidia and the wider lamprocystidia (22–42 × 14–25 µm) [60]; *P. violaceolivida* differs in terms of its pale pink, with a violaceous hymenial surface and a fimbriate margin [60].

In the current study, based on the further ITS phylogenetic tree (Figure 3), two new taxa have been grouped within the genus *Vararia*. These are *V. daweishanensis* and *V. fragilis*, in which *V. daweishanensis* was grouped with four close taxa, namely, *V. gomezii*, *V. rhombospora*, *V. sigmatospora* and *V. trinidadensis,* while *V. fragilis* was grouped with a clade comprising *V. ambigua* and *V. ellipsospora*. However, morphologically speaking, *V. gomezii* differs from *V. daweishanensis* in having a pinkish buff to cream hymenial surface and simple-septate generative hyphae, as well as navicular basidiospores [20]. *V. rhombospora* is separated from *V. daweishanensis* by having a fragile basidiomata with a cream to beige gray hymenial surface, with rhomboid and larger basidiospores (15–17 × 5–6 µm) [61]; *V. sigmatospora* is distinguishable from *V. daweishanensis* by its simple-septate generative hyphae and fusiform, narrower basidiospores (13–15.2 × 2.5–3 µm) [62]; *V. trinidadensis* differs in its gray to grayish-white hymenial surface, simple-septate generative hyphae, and fusiform, narrower basidiospores (13–17 × 2.5–3.2 µm) [63]. *V. ambigua* is distinguishable from *V. fragilis* by its powdery hymenial surface, as well as by basidia that are swollen at the base and larger basidia (27–40 × 3.5–4 µm) [21]; *V. ellipsospora* differs from *V. fragilis* in its fimbriate margin, clamped generative hyphae, wider gloeocystidia (28–48 × 8–11 µm) and larger basidiospores (8–12 × 5.5–6.5 µm) [22].

Morphologically, *Peniophora cinerea* (Pers.) Cooke, *P. laeta* (Fr.) Donk, *P. laurentii* S. Lundell, *P. polygonia* (Pers.) Bourdot & Galzin, *P. rhodocarpa* Rehill & B.K. Bakshi are similar to *P. roseoalba* by having encrusted lamprocystidia. However, *P. cinerea* differs from *P. roseoalba* by its smaller lamprocystidia (15–20 × 6–10 µm), and subcylindrical to allantoid basidiospores [60]; *P. laeta* is separated from *P. roseoalba* by having a hydnoid to raduloid hymenophore, larger gloeocystidia (60–120 × 8–10 µm) and cylindrical to suballantoid, larger basidiospores (9–15 × 3.5–4.5 µm) [64]; *P. laurentii* is distinguished from *P. roseoalba* by tuberculate to plicate or merulioid hymenophore, white margin, simple-septa generative hyphae, as well as longer gloeocystidia (70–150 × 8–12 µm) and larger basidia (50–60 × 6–8 µm) [60]; *P. polygonia* is separated from *P. roseoalba* by having bladder like, bigger gloeocystidia (60–100 × 15–25 µm), presence of dendrohyphidia, and cylindrical to allantoid, larger basidiospores (10–14 × 2.5–4 µm) [60]; *P. rhodocarpa* differs *P. roseoalba* by having tuberculate, rimose hymenial surface, larger gloeocystidia (50–90 × 12–18 µm) with larger lamprocystidia (60–100 × 12–18 µm), and allantoid, narrower basidiospores (5–8.5 × 1.7–2.2 µm) [60].

*Peniophora yunnanensis* is similar to *P. aurantiaca* (Bres.) Höhn. & Litsch., *P. bonariensis* C.E. Gómez, *P. junipericola* J. Erikss., *P. meridionalis* Boidin, *P. quercina* (Pers.) Cooke, based on having clamped generative hyphae and gloeocystidia. However, *Peniophora aurantiaca* is distinguished from *P. yunnanensis* by its orange-red, reddish to reddish grey hymenial surface, larger gloeocystidia (70–150 × 10–20 µm), larger basidia (60–80 × 10–15 µm), and ellipsoid, larger basidiospores (14–20 × 8–12 µm) [60]; *P. bonariensis* can be delimited from *P. yunnanensis* by its pinkish grey to greyish violaceous hymenial surface, thick-walled gloeocystidia and larger lamprocystidia (30–50 × 12–25 µm) [60]; *P. junipericola* differs by having pinkish or greyish red to violaceous hymenial surface, larger lamprocystidia (40–80 × 6–18 µm), and allantoid basidiospores [60]; *P. meridionalis* differs from *P. yunnanensis* by its ochraceous grey, yellowish brown hymenial surface, presence of dendrohyphidia, and larger lamprocystidia (35–55 × 8–20 µm) [60]; *P. quercina* is separated from *P. yunnanensis* by having the pinkish to pinkish grey or bluish grey to violaceous hymenial surface, and larger lamprocystidia (30–80 × 10–20 µm) [60].

*Peniophora yunnanensis* resembles *P. gilbertsonii* Boidin, *P. lilacea* Bourdot & Galzin, *P. limitata* (Chaillet ex Fr.) Cooke, *P. piceae* (Pers.) J. Erikss. and *P. rufomarginata* (Pers.) Bourdot & Galzin in having a tuberculate hymenial surface. However, *Peniophora gilbertsonii* is different from *P. yunnanensis* in having an ochraceous pink to reddish or brown to grey hymenial surface and the presence of dendrohyphidia [60]; *P. lilacea* can be delimited from *P. yunnanensis* along its pinkish grey to ochraceous violaceous hymenial surface, along with its thick-walled gloeocystidia in trauma. We recorded the presence of the dendrohyphidia and ellipsoid, wider basidiospores (9–16 × 6.5–10 µm) [60]; *P. limitata* differs from *P. yunnanensis* by having pinkish gray or violaceous gray to a dark blue-gray hymenial surface, and wider lamprocystidia (25–60 × 8–12 µm) [19]; *P. piceae* is distinguished from *P. yunnanensis* by its reddish grey to grey to a dark violaceous grey hymenial surface, larger lamprocystidia (40–80 × 6–18 µm), and allantoid, narrower basidiospores (6.5–9.5 × 2–2.8 µm) [60]; *P. rufomarginata* is separated from *P. yunnanensis* by having a pinkish to pinkish gray or bluish gray hue to the violaceous hymenial surface, along with larger lamprocystidia (30–80 × 10–20 µm) [60].

*Vararia amphithallica* Boidin, Lanq. & Gilles, *V. bispora* S.L. Liu & S.H. He, *V. montana* S.L. Liu & S.H. He, *V. ochroleuca* (Bourdot & Galzin) Donk and *V. rugosispora* Boidin, Lanq. & Gilles resembles *V. daweishanensis* by having a smooth hymenial surface and clavate to cylindrical basidia. However, *Vararia amphithallica* is distinguished from *V. daweishanensis* by its fimbriate margin, 2-sterigmata basidia, and ellipsoid to cylindrical basidiospores (9–12 × 4–7 µm) [31]; *V. bispora* differs in *V. daweishanensis* by having the thick-walled gloeocystidia, with 2-sterigmata basidia, and larger, fusiform to cylindrical basidiospores (16–24 × 6–8 µm) [31]; *V. montana* is separated from *V. daweishanensis* by having the brittle basidiomata, longer gloeocystidia (50–100 × 4–9 µm), and broadly ellipsoid, larger basidiospores (16–24 × 8–14 µm) [31]; *V. ochroleuca* differs from *V. daweishanensis* by having cream-colored to pallid ochraceous hymenial surface, slightly thick-walled gloeocystidia, simple-septa generative hyphae, and broadly ellipsoid, to drop-shaped, smaller basidiospores (2.6–3.8 × 2–3.2 µm) [65]. *V. rugosispora* can be delimited from *V. daweishanensis* by its simple-septate generative hyphae and longer basidiospores (12–16 × 7–8 µm) [21].

*Vararia breviphysa* Boidin & Lanq., *V. cinnamomea* Boidin, Lanq. & Gilles, *V. cremea* Boidin, Lanq. & Gilles, *V. gallica* (Bourdot & Galzin) Boidin, *V. hauerslevii* Boidin, and *V. sinapicolor* Boidin & Gilles are similar to *V. fragilis*, based on characteristics such as the thick-walled dichohyphae, and four sterigmata basidia. However, *V. breviphysa* differs from *V. fragilis* by having the larger gloeocystidia (50–65 × 6–8.5 µm), fusiform and larger basidiospores (15–22 × 4–6 µm) [20]. *V. cinnamomea* is distinguished from *V. fragilis* by its cinnamon hymenial surface, larger basidia (45–65 × 8–10 µm), and larger basidiospores (9–13 × 5–7.2 µm) [25]. *V. cremea* can be delimited from *V. fragilis* by the longer gloeocystidia (40–90 × 7–15 µm), and larger basidiospores (15–20 × 2.7–3.5µm) [21]. *V. gallica* differs from *V. fragilis* in having a whitish hymenial surface and larger basidiospores (9–12 × 3.5–5 µm) [19]. *V. hauerslevii* is separated from *V. fragilis* by its larger gloeocystidia (50–60 × 7–9 µm) and subfusoid, larger basidiospores (10–15 × 3.5–4.5 µm) [66]. *V. sphaericospora* differs from *V. fragilis* in having clamped generative hyphae, bigger basidia (33–45 × 6–7 µm), and larger basidiospores (12.5–14 × 5.2–7 µm) [20,21,23].

The taxa of *Peniophora* and *Vararia* are typical examples of wood-rotting fungi, which is an extensively studied family [19,67,68,69,70]. So far, several studies on new wood-decaying fungi belonging to the *Peniophora* and *Vararia* from China have been reported [34,71,72,73,74,75].

## Figures and Tables

**Figure 1 jof-08-01227-f001:**
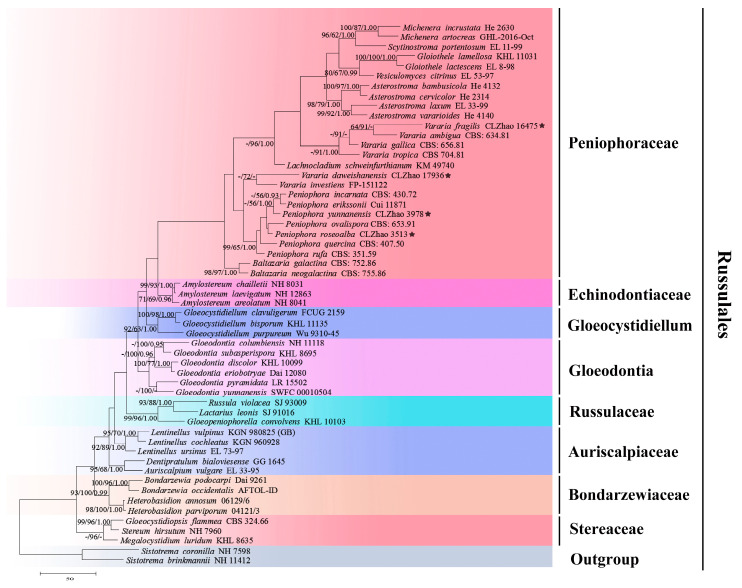
A maximum parsimony strict consensus tree, illustrating the phylogeny of four new species and related genera in the order Russulales, based on nLSU sequences. The branches are labeled with maximum likelihood bootstrap values of >70%, parsimony bootstrap values of >50%, and Bayesian posterior probabilities of >0.95, respectively. The new species are marked with asterisks.

**Figure 2 jof-08-01227-f002:**
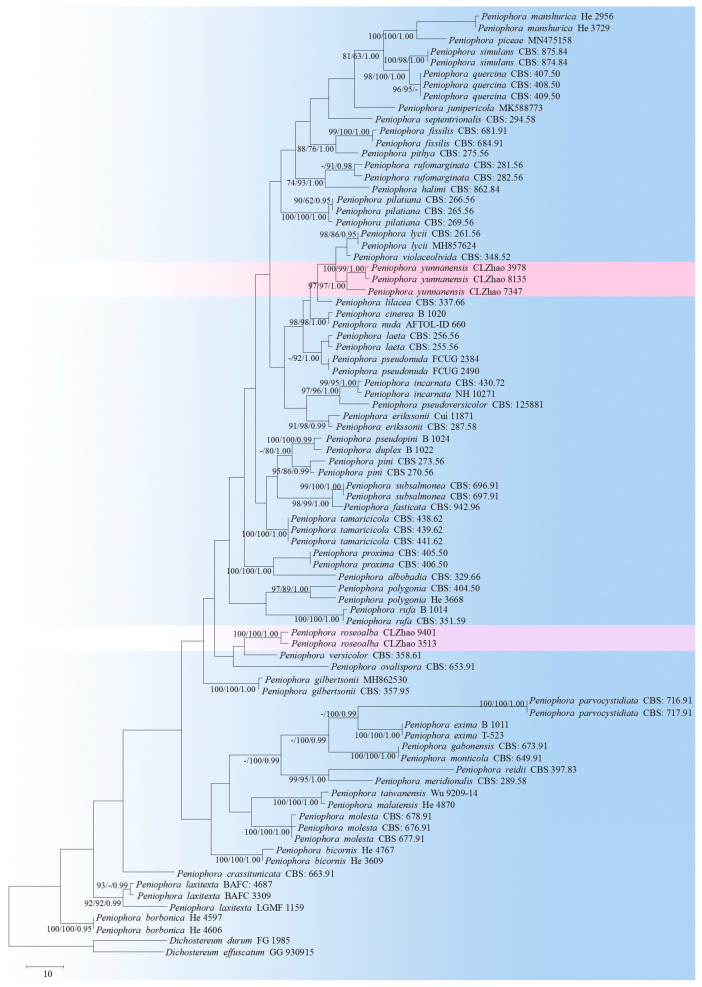
A maximum parsimony strict consensus tree, illustrating the phylogeny of two new species and the related species in the genus *Peniophora*, based on the ITS sequences. The branches are labeled with maximum likelihood bootstrap values higher than 70%, parsimony bootstrap proportions that are higher than 50%, and Bayesian posterior probabilities of more than 0.95, respectively.

**Figure 3 jof-08-01227-f003:**
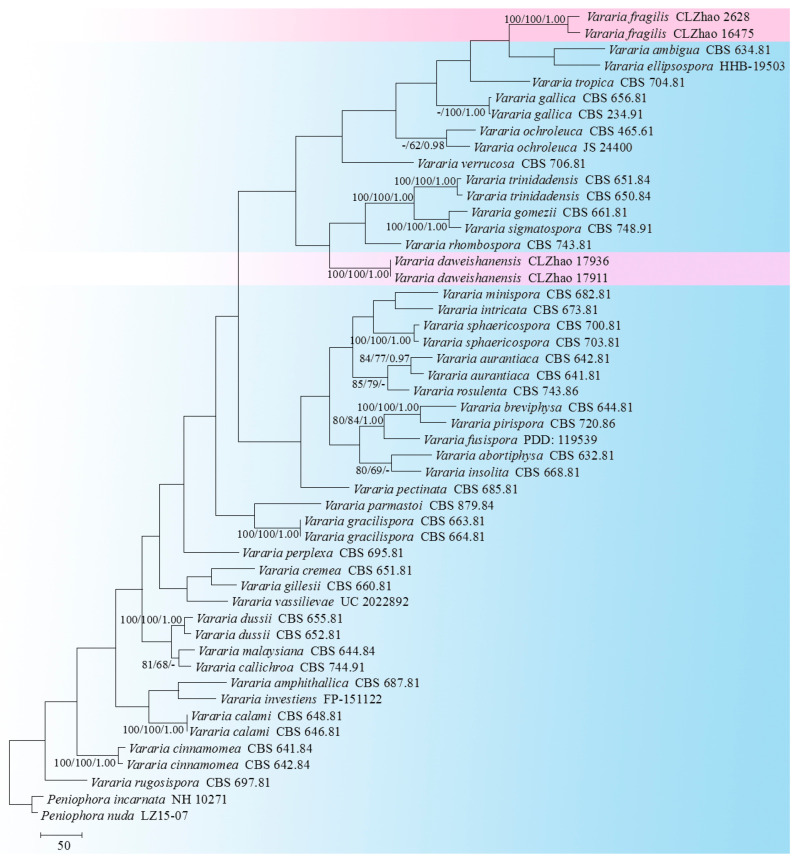
A maximum parsimony strict consensus tree illustrating the phylogeny of two new species and related species in the genus *Vararia*, based on ITS sequences. The branches are labeled with a maximum likelihood bootstrap value of >70%, a parsimony bootstrap value of >50%, and Bayesian posterior probabilities of >0.95, respectively.

**Figure 4 jof-08-01227-f004:**
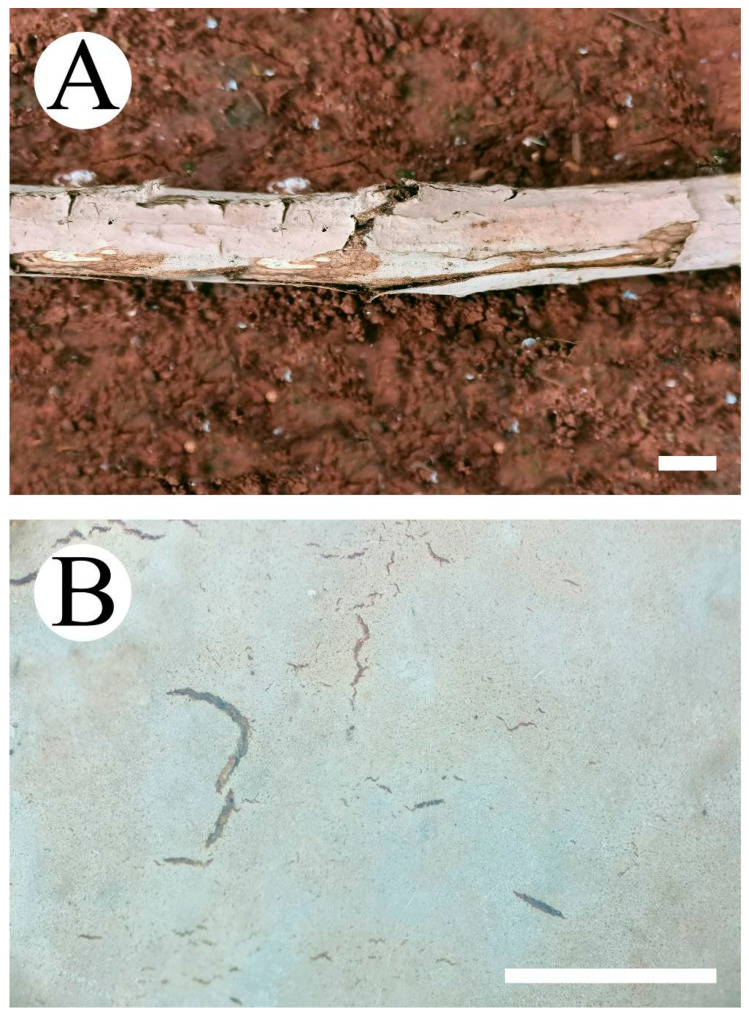
Basidiomata of *Peniophora roseoalba* (holotype): the front of the basidiome (**A**); the characteristic hymenophore (**B**). Bars: (**A**) = 1 cm and (**B**) = 1 mm.

**Figure 5 jof-08-01227-f005:**
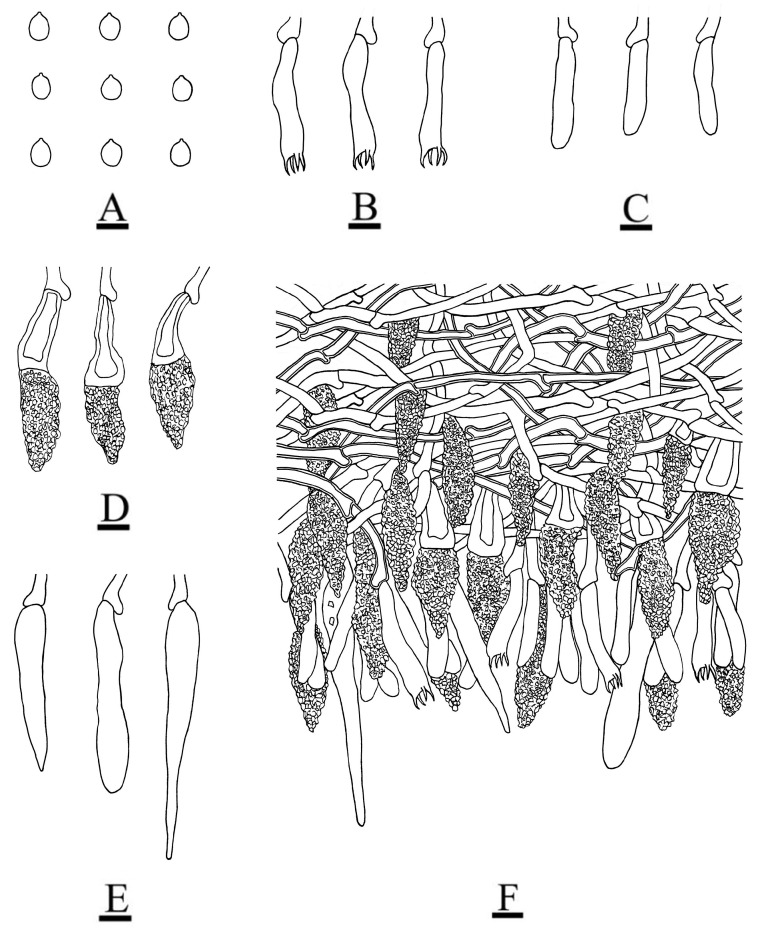
Microscopic structures of the *Peniophora roseoalba* (holotype): basidiospores (**A**); basidia (**B**); basidioles (**C**); lamprocystidia (**D**); subcylindrical to conical gloeocystidia (**E**); a section of the hymenium (**F**). Bars: (**A**–**F**) = 10 µm.

**Figure 6 jof-08-01227-f006:**
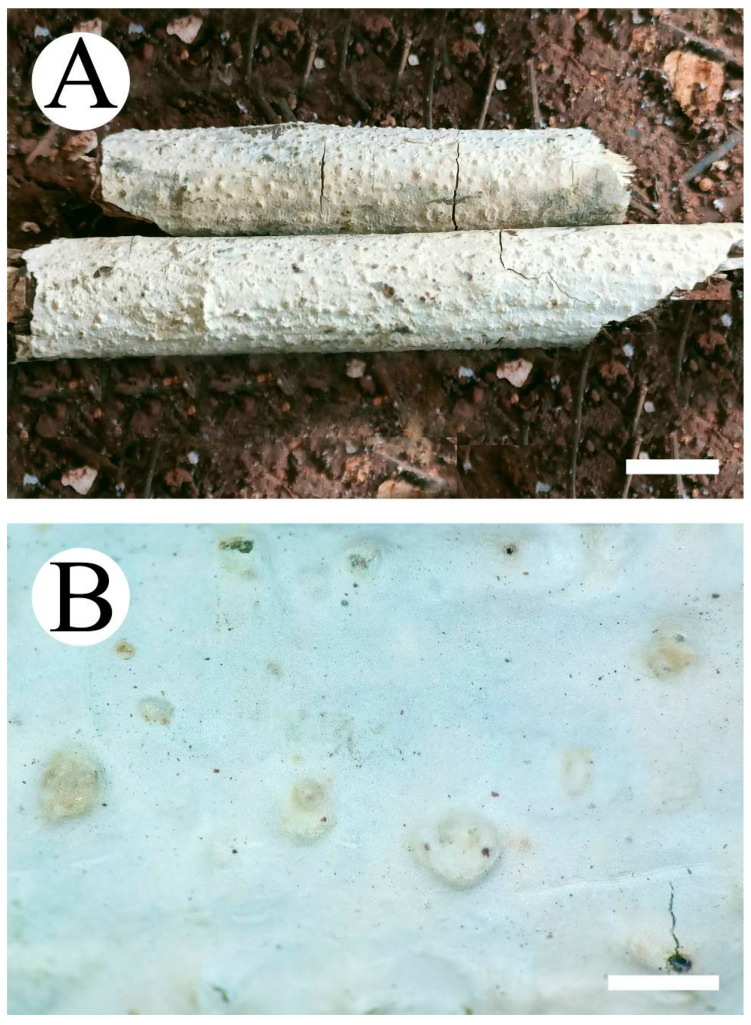
Basidiomata of *Peniophora yunnanensis* (holotype): the front of the basidiomata (**A**); the characteristic hymenophore (**B**). Bars: (**A**) = 1 cm and (**B**) = 1 mm.

**Figure 7 jof-08-01227-f007:**
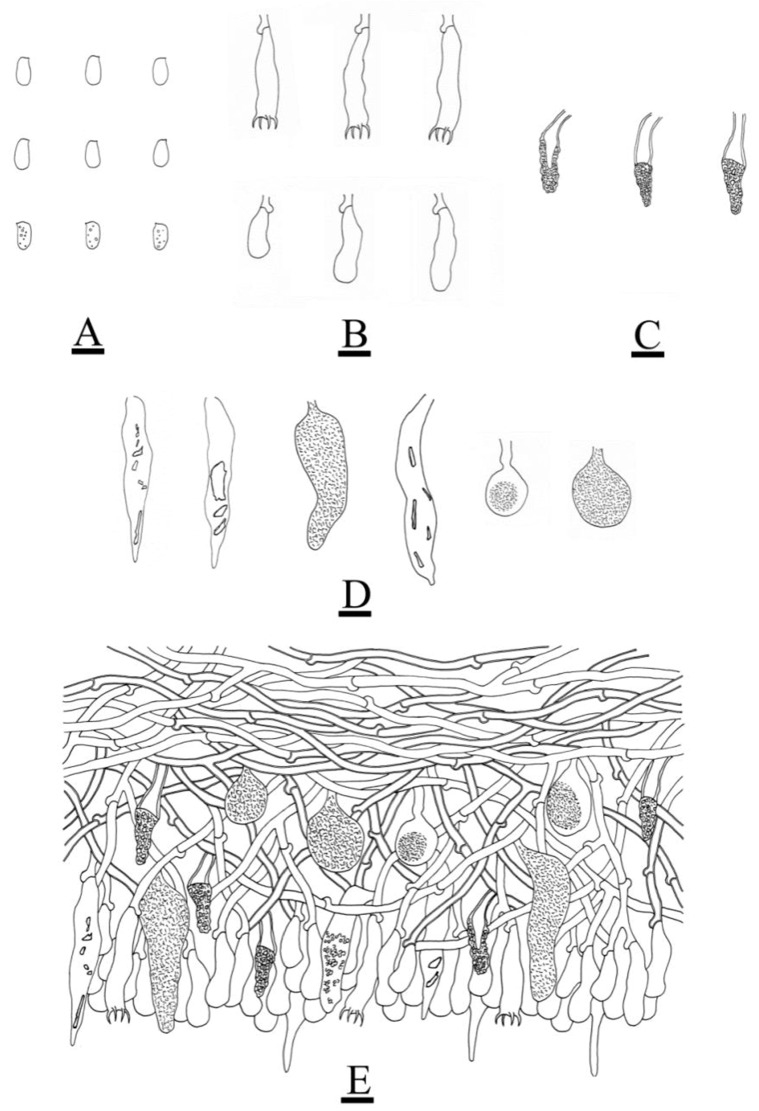
Microscopic structures of *Peniophora yunnanensis* (holotype): basidiospores (**A**); basidia and basidioles (**B**); lamprocystidia (**C**); the conical, clavate to fusiform, subglobose gloeocystidia (**D**); a section of the hymenium (**E**). Bars: (**A**–**E**) = 10 µm.

**Figure 8 jof-08-01227-f008:**
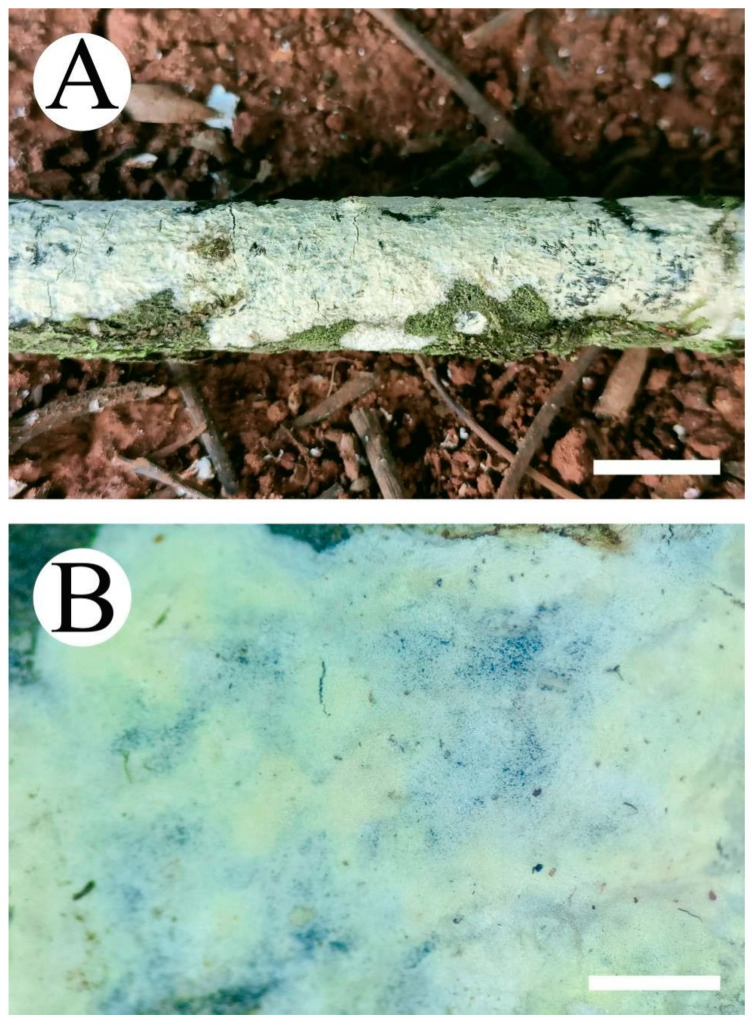
Basidiomata of the *Vararia daweishanensis* (holotype): the front of the basidiomata (**A**); the characteristic hymenophore (**B**). Bars: (**A**) = 1 cm and (**B**) = 1 mm.

**Figure 9 jof-08-01227-f009:**
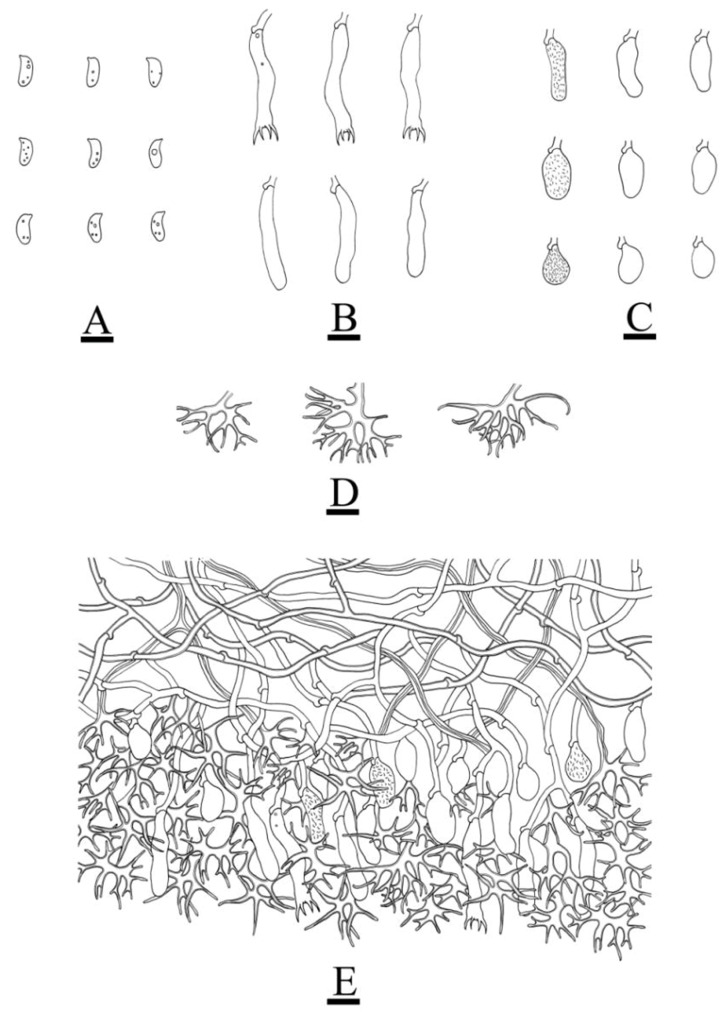
Microscopic structures of *Vararia daweishanensis* (holotype): basidiospores (**A**); basidia and basidioles (**B**); subcylindrical, elliptical to ovoid gloeocystidia (**C**); dichohyphae (**D**); a section of the hymenium (**E**). Bars: (**A**–**E**) = 10 µm.

**Figure 10 jof-08-01227-f010:**
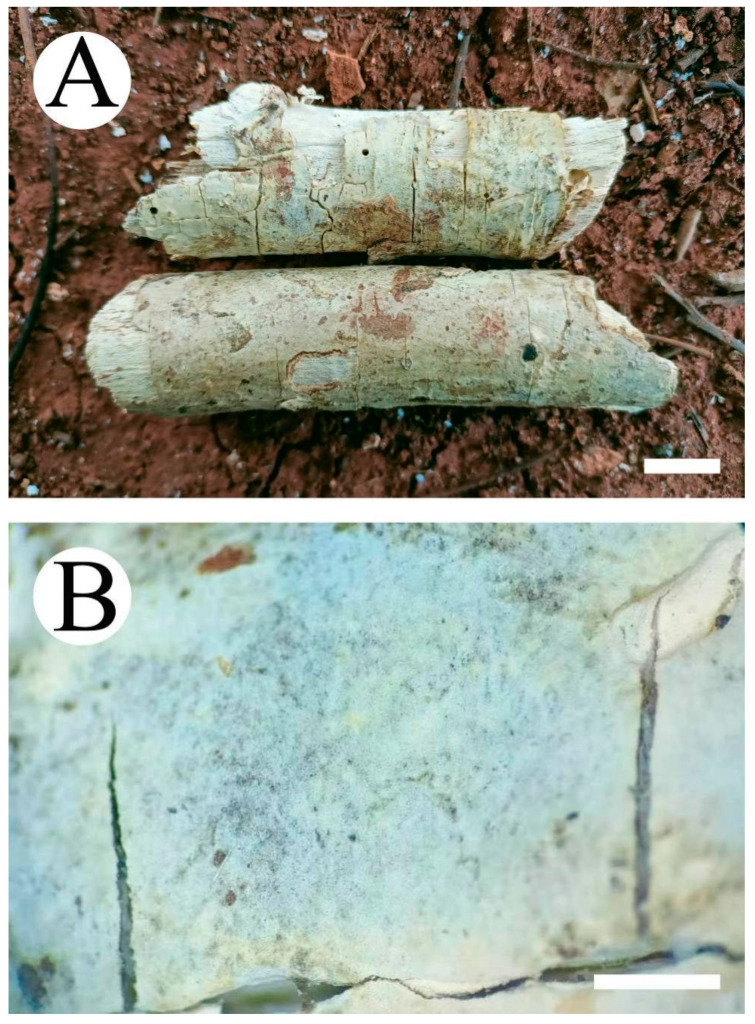
Basidiomata of *Vararia fragilis* (holotype): the front of the basidiomata (**A**); the characteristic hymenophore (**B**). Bars: (**A**) = 1 cm and (**B**) = 1 mm.

**Figure 11 jof-08-01227-f011:**
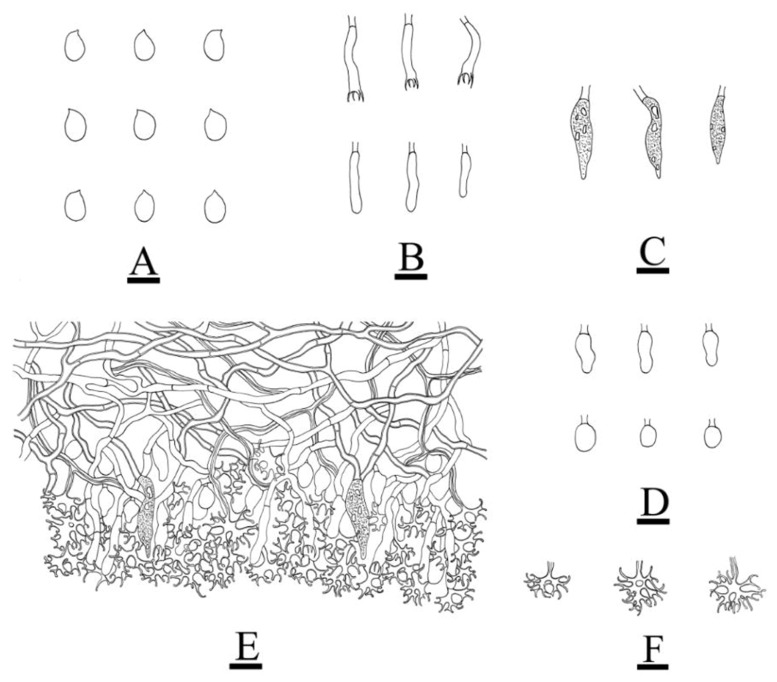
Microscopic structures of *Vararia fragilis* (holotype): basidiospores (**A**); basidia and basidioles (**B**); fusiform gloeocystidia (**C**); elliptical to ovoid gloeocystidia (**D**); a section of the hymenium (**E**); dichohyphae (**F**). Bars: (**A**) = 5 μm, (**B**–**F**) = 10 µm.

## Data Availability

Publicly available datasets were analyzed in this study. This data can be found here: [https://www.ncbi.nlm.nih.gov/; https://www.mycobank.org/page/Simple%20names%20search; http://purl.org/phylo/treebase, submission ID 28664; all accessed on 13 October 2022].

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
