# Peer review of "Four New Wood-Inhabiting Fungal Species of Peniophoraceae (Russulales, Basidiomycota) from the Yunnan-Guizhou Plateau, China"

_jof, 2022, doi:10.3390/jof8111227_

Round 1
Reviewer 1 Report
This is a taxonomic study centered in the description of 4 new species. I recommend you to improve the English and to focus on the taxonomy; avoid anything else as the relationship with insects, wood-decay and so on: they are not related to the aim of your study.

Reviewer 2 Report
Dear Editor
After reviewing the manuscript “Phylogenetic and Taxonomic Analyses of Four New Wood-Inhabiting Fungi of Peniophoraceae (Russulales, Basidiomycota) from the Yunnan-Guizhou Plateau”, I found that it can be published after attending the suggestions made and indicated directly in the text, they are minor observations with the aim of improvising it. The main contribution is the discovery of new species but it does require in some parts to improve the wording, each suggestion is clearly indicated in the text.
